# Causally Steered Diffusion for Video Counterfactual Generation

Figure 1: **Generated video counterfactual results**: Our CSVC framework leverages LLMs to transform factual parent prompts into counterfactual ones, while LDM-based editing systems model the mapping from a factual video to its counterfactual version given the transformed parents. We illustrate interventions on age (e.g., making a woman appear young) and gender (e.g., transforming a woman into a man with a beard). Within CSVC, the VLM-based textual loss improves counterfactual effectiveness (third row) by steering the generation process through causal refinement of the counterfactual parent prompt.

## Abstract

Adapting text-to-image (T2I) latent diffusion models (LDMs) to video editing has shown strong visual fidelity and controllability, but challenges remain in maintaining causal relationships inherent to the video data generating process. In this work, we propose CSVC, a framework for counterfactual video generation grounded in structural causal models (SCMs) and formulated as an out-of-distribution (OOD) prediction task. CSVC builds on black-box counterfactual functions, which approximate SCM mechanisms without explicit structural equations. In our framework, large language models (LLMs) generate counterfactual prompts that are consistent with a predefined causal graph, while LDM-based video editors produce the corresponding video counterfactuals. To ensure faithful interventions, we introduce a vision–language model (VLM)-based textual loss that refines prompts to enforce counterfactual conditioning, steering the LDM latent space toward causally meaningful OOD variations without internal model access or fine-tuning. Experiments on real-world facial videos show that CSVC achieves state-of-the-art causal effectiveness while preserving temporal consistency and visual quality. By combining SCM reasoning with black-box generative models, CSVC enables

realistic "what if" hypothetical video scenarios with applications in digital media and healthcare.

# 1 INTRODUCTION

Text-to-image (T2I) latent diffusion models (LDMs) have significantly advanced the field of image generation Podell et al. (2024); Rombach et al. (2022), showcasing remarkable fidelity and enhanced creative control in image editing Cong et al. (2024); Feng et al. (2024); Geyer et al. (2024); Jeong & Ye (2024); Kara et al. (2024). However, the efficacy of image editing is not consistent, as modifications affecting attributes with causal dependencies often generate unrealistic and potentially misleading results if these relationships are disregarded. This issue is particularly critical in data where causal interplays determine the imaging content Melistas et al. (2024); Pawlowski et al. (2020); Papanastasiou et al. (2024).

Recent efforts in video editing adapt T2I models to address the challenge of maintaining spatiotemporal consistency Gu et al. (2024); Liu et al. (2024a); Shin et al. (2024); Zhang et al. (2023); Zhao et al. (2024); Cong et al. (2024); Geyer et al. (2024); Wu et al. (2023b). Some approaches achieve text-guided editing by fine-tuning pre-trained models Gu et al. (2024); Shin et al. (2024); Zhang et al. (2023); Zhao et al. (2024), whereas others enable zero-shot Cong et al. (2024); Geyer et al. (2024) or one-shot Wu et al. (2023b); Liu et al. (2024a) editing with minimal training overhead.

Yet, in contrast to these developments, existing video-editing methods overlook predefined causal graphs and Pearl-style counterfactual reasoning Pearl (2009). SCMs encode causal relations as directed acyclic graphs (DAGs) with mechanisms relating variables to their direct causes (termed parents) and unobserved exogenous factors. In high-dimensional domains, deep generative models can approximate these mechanisms, but their lack of identifiability Locatello et al. (2020); Khemakhem et al. (2020) often entangles causal effects Pawlowski (2022), hindering recovery of the true causal structure. Monteiro et al. (2023) conceptualize SCM mechanisms as black-box counterfactual functions, formulating them as functional assignments that map factual observations and intervened parent variables to counterfactual outcomes.

We propose to operationalize SCM counterfactuals for high-dimensional video variables by implementing counterfactual functions Monteiro et al. (2023) as black-box generative AI models, where counterfactual conditioning (intervened parents) is expressed through natural-language prompts. Large language models (LLMs) are employed to model parent variables, translating factual prompts into counterfactual descriptions aligned with a causal DAG. In parallel, text-guided LDM-based video editing systems implement the black-box counterfactual function of the video variable.

Moreover, previous work Ribeiro et al. (2023); Chen et al. (2016) shows that generative models often disregard counterfactual conditioning, so outputs may fail to reflect the intended interventions. Inspired by these findings, we build on the hypothesis that pre-trained LDMs already encode plausible causal counterfactuals within their learned distribution. To realize them, we introduce a vision–language model (VLM) textual loss that enforces target counterfactual parents (prompts) and steers the latent space of the LDM toward generating out-of-distribution (OOD) samples consistent with these interventions. We argue that refining parent textual prompts via the proposed textual loss provides an implicit yet powerful mechanism for steering generation toward effective and realistic counterfactual estimations, while operating entirely in a black-box setting. This stands in contrast to approaches based on attention engineering Geyer et al. (2024); Qi et al. (2023); Cong et al. (2024); Wang et al. (2025), which offer suboptimal solutions and require access to model internals.

This paper proposes "Causal Steering for Video Counterfactuals" (CSVC), a framework for counterfactual video generation conceptualized as structured OOD generation. CSVC encodes predefined causal relationships into text prompts representing parent variables and leverages black-box generative AI models to implement counterfactual mappings for both prompts and videos. To enforce causal consistency, it incorporates a VLM-based textual loss that refines parent prompts via textual differentiation, guiding LDM-based editors toward causally faithful edits without weight updates or feature engineering. Our objective is to modify attributes of a factual video while ensuring semantic coherence and causal alignment. To this end, LLMs generate causally consistent counterfactual prompts, LDMs produce the corresponding video edits, and the VLM loss steers the diffusion latent space toward interventions consistent with the causal graph. As shown in Figure 1, refining par-

ent prompts to enforce causal constraints (interventions) improves counterfactual fidelity, enabling diverse and realistic OOD counterfactuals aligned with the intended interventions.

In summary, our contributions are:

- We propose the first framework (CSVC) for implementing SCM-style video counterfactuals by leveraging generative AI models as black-box counterfactual functions. CSVC operates entirely in a black-box setting, requiring no access to the internal parameters of the generative AI models used for the counterfactual mappings.
- CSVC introduces a VLM-based textual loss that enforces counterfactual conditioning by refining parent prompts through propagated textual gradients, thereby steering the latent space of LDMs toward semantically meaningful and causally consistent counterfactuals.
- Our approach achieves state-of-the-art causal effectiveness on diverse real-world facial videos across multiple interventions (e.g., age, gender, beard, baldness) while preserving video quality, minimality, and temporal coherence.
- We design novel VLM-based metrics to assess causal effectiveness and minimality, offering interpretable and scalable evaluation tools for counterfactual video generation.

## 2 RELATED WORK

**Latent Diffusion-based Video Editing.** LDMs Podell et al. (2024); Rombach et al. (2022) have driven major progress in video generation and editing Croitoru et al. (2023); Sun et al. (2024). Existing approaches include tuning-based methods that adapt text-to-image or text-to-video models via cross-frame attention or few-shot fine-tuning Podell et al. (2023); Zhang et al. (2023); Wu et al. (2023b); Liu et al. (2024a); Shin et al. (2024); Gu et al. (2024); Wang et al. (2025); Zhao et al. (2024); controlled editing methods such as ControlNet Chen et al. (2023), which leverage priors like optical flow, depth, or pose Yang et al. (2023); Hu & Xu (2023); Feng et al. (2024); Ma et al. (2024); Yang et al. (2025); and training-free methods that exploit diffusion features, latent fusion, noise shuffling, or optical-flow guidance Tang et al. (2023); Qi et al. (2023); Khandelwal (2023); Kara et al. (2024); Chu et al. (2024); Cong et al. (2024); Yang et al. (2024); Jeong & Ye (2024). In our framework, we adopt lightweight one-shot and zero-shot T2I LDM-based video editing methods to model the counterfactual mapping from a source video to an edited one, conditioned on a parent text prompt describing the interventions.

**Counterfactual Image and Video Generation.** Visual counterfactual generation explore hypothetical "what-if" scenarios through targeted and semantically meaningful modifications to the input Wachter et al. (2017); Schölkopf et al. (2021). It is applied in counterfactual explainability Verma et al. (2024); Augustin et al. (2022); Jeanneret et al. (2022; 2023); Weng et al. (2024); Pegios et al. (2024b;a); Sobieski et al. (2025), robustness testing Dash et al. (2022); Prabhu et al. (2023); Le et al. (2023); Lai et al. (2024); Yu & Li (2024); Zhang et al. (2024); Weng et al. (2024), and causal inference Pearl (2009); Vlontzos et al. (2022; 2023; 2025); Pawlowski et al. (2020); Kocaoglu et al. (2018); Xia et al. (2021); Abdulaal et al. (2022); Sanchez & Tsaftaris (2022); Ribeiro et al. (2023); Sanchez et al. (2022); Fontanella et al. (2024); Song et al. (2024). While much work focuses on static images Monteiro et al. (2023); Ribeiro et al. (2023); Melistas et al. (2024), the temporal coherence of causal counterfactual video generation remains underexplored Reynaud et al. (2022). In contrast to prior work, we introduce an SCM-faithful framework for video counterfactuals by approximating causal mechanisms with generative AI models under the black-box counterfactual functions approach.

**Evaluation of Visual Editing and Counterfactuals.** Evaluating counterfactuals is inherently challenging Schölkopf et al. (2021); Melistas et al. (2024). Standard metrics assess image quality Korhonen & You (2012); Zhang et al. (2018a); Wang et al. (2004); Heusel et al. (2017) and semantic alignment Radford et al. (2021a), but causal counterfactuals Melistas et al. (2024); Galles & Pearl (1998); Halpern (2000) require stricter criteria, such as causal effectiveness Monteiro et al. (2023) and minimality Sanchez & Tsaftaris (2022). In video, evaluation is further complicated by the need for temporal consistency, while existing benchmarks Liu et al. (2023); Yuan et al. (2024); Liu et al. (2024b); Huang et al. (2024); Ji et al. (2024); Sun et al. (2024) largely overlook counterfactual reasoning. Additionally, widely used video metrics such as DOVER Wu et al. (2023a), CLIP Score Radford et al. (2021a), and flow warping error Lai et al. (2018) fail to capture causal relationships.

To address this, we evaluate generated counterfactual videos using both causal adherence–via counterfactual effectiveness and minimality Monteiro et al. (2023); Ribeiro et al. (2023); Melistas et al. (2024)–and overall video quality and temporal consistency. For minimality, we introduce a novel VLM-based metric, enabling comprehensive assessment of causal fidelity in text-guided video counterfactual generation.

## 3 BACKGROUND

**T2I LDMs for Video Editing.** Recent text-guided video editing methods Wu et al. (2023b); Cong et al. (2024); Geyer et al. (2024) employ pre-trained T2I LDMs, typically Stable Diffusion Rombach et al. (2022), that operate on a latent image space. A pre-trained autoencoder $(\mathcal{E}, \mathcal{D})$ Kingma et al. (2013); Van Den Oord et al. (2017) maps an image frame $x$ to a latent code $z = \mathcal{E}(x)$, with $\mathcal{D}(z) \approx x$. A conditional U-Net Ronneberger et al. (2015) denoiser $\epsilon_\theta$ is trained to predict noise in the latent $z_t$ at diffusion timestep $t$, minimizing:

$$E_{z, \epsilon \sim \mathcal{N}(0,1),\, t,\, c}\big[\|\epsilon - \epsilon_\theta(z_t, t, c)\|_2^2\big],$$

where $c$ is the embedding of text prompt $\mathcal{P}$. The U-Net $\epsilon_\theta$ can be either inflated into a 3D spatio-temporal network for one-shot video fine-tuning Wu et al. (2023b) and zero-shot optical-flow guidance Cong et al. (2024), or directly used for frame editing, with temporal consistency imposed via feature propagation Geyer et al. (2024). These methods leverage deterministic DDIM Song et al. (2021) sampling and inversion which allows to reconstruct or edit the original video frames. Although each method has its own temporal regularization strategies and heuristics, given an input video $\mathcal{V}$ and an editing prompt $\mathcal{P}$, the core video editing process can be expressed as:

$$\mathcal{V}' = \mathcal{D}(DDIM\text{-}sampling(DDIM\text{-}inversion(\mathcal{E}(\mathcal{V})), \mathcal{P})) \tag{1}$$

**Causal Framework for Video Counterfactuals.** A Structural Causal Model (SCM) Pearl (2009) represents a system as a set of functional assignments, where each variable is determined by its direct causes (termed parents) and an exogenous noise term. Within this framework, counterfactual inference follows the abduction-action-prediction paradigm. Mapping this to diffusion-based video editing, DDIM inversion corresponds to *abduction* (inferring the exogenous noise $\epsilon$), the *action* step is the prompt-based intervention using the editing prompt $\mathcal{P}$, and DDIM sampling performs the *prediction*, producing the counterfactual video $\mathcal{V}'$.

**Counterfactual Functions as Black-Box Mechanisms.** While Pearl's SCM-based formulation provides a principled view of counterfactuals, applying structural equations or inferring the exogenous noise $\epsilon$ in high-dimensional domains such as video is often intractable Locatello et al. (2020); Khemakhem et al. (2020). Following Monteiro et al. (2023), we instead adopt black-box counterfactual functions, where a counterfactual outcome $x'$ is obtained as: $x' = f(x, pa, pa')$, with $x$ the factual observation, $pa$ its factual parents (direct causes), and $pa'$ the intervened parents. Here, $f$ is treated as an opaque mechanism that subsumes abduction, action, and prediction.

## 4 METHODOLOGY

We build our framework on black-box counterfactual functions, which model counterfactual outcomes as mappings from factual inputs and counterfactual parents (interventions) without requiring explicit structural equations or abduction of the exogenous noise $\epsilon$.

### 4.1 CAUSAL STEERING FOR VIDEO COUNTERFACTUALS (CSVC)

**LLMs as Black-Box Counterfactual Functions for Parent Variables.** Causal knowledge can be injected into video editing systems through target prompts that encode the relationships of a DAG $\mathcal{G}$. We assume a target prompt $P$ represents the counterfactual parents $pa'$ of the video variable $\mathcal{V}$ (Figure 2). Following the black-box counterfactual framework of Monteiro et al. (2023), we use LLMs to generate counterfactual prompts. As shown in Figure 2, the LLM receives the factual prompt $P_{factual}$, the causal graph $\mathcal{G}$, and an in-context learning (ICL) prompt $P_{ICL}$ (Appendix A.4.1) with factual–counterfactual examples. The graph specifies which relations to preserve, while the ICL prompt guides the mapping to valid counterfactual prompts $P$ (parents). Formally,

$$P = g_{\text{LLM}}(P_{factual}, \mathcal{G}, P_{ICL}) \tag{2}$$

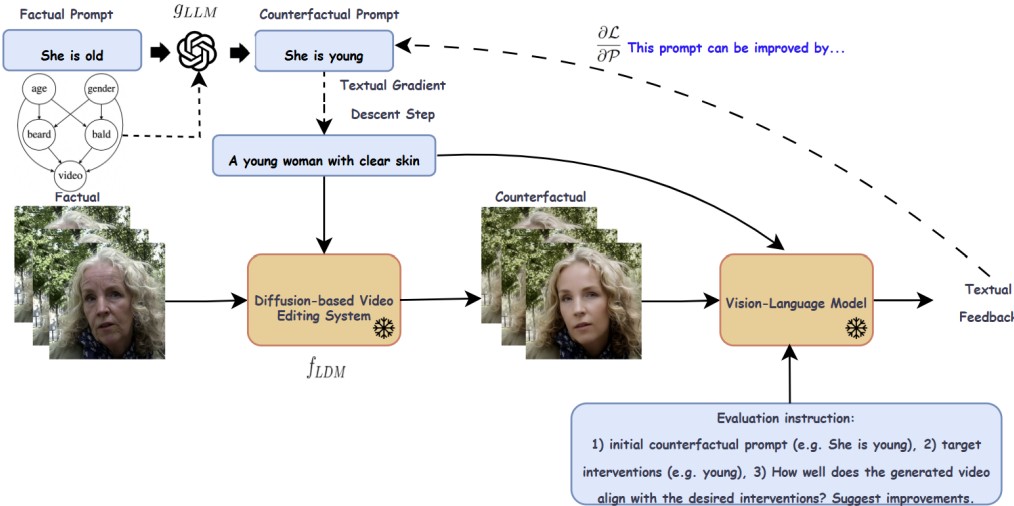

Figure 2: **CSVC at a glance:** The initial counterfactual prompts (e.g., She is young) are generated using an LLM black-box counterfactual function $g_{LLM}$ by providing the causal graph and the factual prompts (e.g., She is old) and leveraging in-context learning Dong et al. (2022). The video editing system operates as a black-box $f_{LDM}$ (frozen) counterfactual generator and the (black-box) VLM as an evaluator of the generated counterfactuals which implements our proposed textual loss. The VLM takes as input a generated counterfactual frame, the evaluation instruction, and the target counterfactual prompt $\mathcal{P}$, and outputs textual feedback used to compute a "textual" gradient $\frac{\partial \mathcal{L}}{\partial \mathcal{P}}$, thereby optimizing the textual loss by refining $\mathcal{P}$ and focusing on unsuccessful parent interventions.

**Video Editing Systems as Black-Box Counterfactual Functions for Video Variable.** To model the mechanism that maps factual video observations $\mathcal{V}$ to counterfactual outcomes $\mathcal{V}'$ given a counterfactual prompt $P$ (parents), we employ LDM-based video editing systems. We treat the editing method as an opaque black-box function for counterfactual generation (Figure 2), assuming no access to the $\epsilon_\theta$ LDM parameters (i.e., no updates or backpropagation) and no control over internal processes such as DDIM sampling or inversion. For any prompt-based video editing system $f_{LDM}$, with input video $\mathcal{V}$ and counterfactual parent prompt $\mathcal{P}$, Equation 1 becomes:

$$\mathcal{V}' = f_{LDM}(\mathcal{V}, \mathcal{P}). \tag{3}$$

Our CSVC framework is compatible with any black-box, text-guided diffusion video editing system and is evaluated with three such methods.

**VLM-based textual loss for steering counterfactual video generation.** We observe that the LDM backbones often ignore counterfactual conditioning $P$, failing to incorporate target interventions. To address this, we build on the hypothesis that causally faithful counterfactuals reside within the learned space of LDMs and introduce a VLM-based textual loss designed to enforce the counterfactual parents. The proposed loss steers generation toward effective OOD video counterfactuals by refining only the target parent prompts, without requiring access to the internal parameters of $f_{LDM}$.

To optimize the proposed loss, we employ TextGrad Yuksekgonul et al. (2025), which naturally enables optimization of textual losses. In particular, we perform prompt-level causal steering by refining counterfactual prompts based on the underlying causal relationships and target interventions. TextGrad leverages LLMs to generate natural-language "textual gradients," which are used for iterative refinement of complex systems through textual feedback. Building on this, we design a counterfactual "multimodal loss" with a VLM to guide video generation towards the target interventions. Given a generated counterfactual video frame, the counterfactual parent prompt, and an evaluation instruction containing the target interventions, we implement our proposed "multimodal loss" using a VLM:

$$\mathcal{L} = VLM(\mathcal{V}'_{frame}, evaluation\ instruction, \mathcal{P}), \tag{4}$$

where the evaluation instruction (Appendix A.4.2) is a well-defined textual input to the VLM to suggest improvements on $\mathcal{P}$ based on how well the generated visual input $\mathcal{V}'_{frame}$ (extracted from

$\mathcal{V}'$) aligns with the target counterfactual parents. We further augment the *evaluation instruction* with a causal decoupling (Appendix A.4.3) text input that instructs the VLM to ignore *upstream* variables when intervening on *downstream* ones. This yields optimized prompts that omit explicit upstream references (e.g., neutralizing gender), enabling the LDM backbone to generate samples that intentionally violate the causal graph, such as rendering a woman with a beard (Figure 3). We employ *Textual Gradient Descent* (TGD) Yuksekgonul et al. (2025) to optimize the proposed loss by directly updating the counterfactual parent prompt $\mathcal{P}$:

$$\mathcal{P}' = \text{TGD.step}\left(\mathcal{P}, \frac{\partial \mathcal{L}}{\partial \mathcal{P}}\right)$$

$$= LLM\left(Criticisms\ on\ \{\mathcal{P}\} : \left\{\frac{\partial \mathcal{L}}{\partial \mathcal{P}}\right\}, Incorporate\ the\ criticisms\ and\ produce\ a\ new\ prompt.\right) \tag{5}$$

where $\frac{\partial \mathcal{L}}{\partial \mathcal{P}}$ [1] denotes the "textual gradients", passed through an *LLM* [2] at each TGD update to generate a new counterfactual parent prompt incorporating the VLM criticisms. Optimization halts when the target interventions are met or the maximum number of iterations is reached. The proposed CSVC framework is summarized in Figure 2 and Algorithm 1.

---

**Algorithm 1** Causal Steering for Video Counterfactuals (CSVC)

---

**Require:** Factual prompt $\mathcal{P}_{factual}$, DAG $\mathcal{G}$, ICL prompt $P_{ICL}$, LLM $g_{LLM}$, factual video $\mathcal{V}$, DiffusionVideoEditor $f_{LDM}$, *VLM*
**Ensure:** Counterfactual video $\mathcal{V}'$
1: $P \leftarrow g_{LLM}(P_{factual}, \mathcal{G}, P_{ICL})$ ▷ Counterfactual function for prompt variable (Eq. 2)
2: $prompt \leftarrow \mathcal{P}$ ▷ Initialize counterfactual parent prompt
3: $optimizer \leftarrow \text{TGD}(\text{parameters} = [prompt])$ ▷ Set up textual optimizer
4: **for** $iter = 1$ to $maxIters$ **do**
5: $\quad \mathcal{V}' \leftarrow f_{LDM}(\mathcal{V}, prompt)$ ▷ Counterfactual function of video variable(Eq.3)
6: $\quad loss \leftarrow VLM(\mathcal{V}'_{frame_i}, evaluation\ instruction, prompt)$ ▷ Counterfactual textual loss (Eq. 4
7: $\quad$ **if** "no optimization" $\in$ loss.value **then**
8: $\quad\quad$ **break**
9: $\quad$ **end if**
10: $\quad loss.\text{backward}()$ ▷ Computation of $\frac{\partial \mathcal{L}}{\partial \mathcal{P}}$
11: $\quad optimizer.\text{step}()$ ▷ Update prompt via TGD Eq. (5)
12: **end for**
13: **return** Final counterfactual video $\mathcal{V}'$

---

### 4.2 VLMs for assessing causal effectiveness

Effectiveness is key in counterfactual generation, indicating if the target intervention succeeded Galles & Pearl (1998); Monteiro et al. (2023); Melistas et al. (2024). CLIP-based metrics Radford et al. (2021b) lack interpretability and are inefficient for capturing *causal* alignment between text and image. Following (Hu et al., 2023), we use a VLM to assess effectiveness across a set of generated counterfactual videos with a visual question answering (VQA) approach. Given triplets $\{Q_i^\alpha, C_i, V'_{frame_i}\}_{i=1}^N$, where $Q_i^\alpha$ is a multiple-choice question about the intervened attribute $\alpha$, $C_i$ is the correct answer extracted from the target counterfactual prompt, and $\mathcal{V}'_{frame_i}$ is a generated counterfactual video frame, we measure effectiveness by the accuracy of the VLM's answer:

$$Effectiveness(\alpha) = \frac{1}{N} \sum_{i=1}^N \mathbf{1} \left[VLM(\mathcal{V}'_{frame_i}, Q_i^\alpha) = C_i\right]. \tag{6}$$

### 4.3 VLMs for assessing minimality

Minimal interventions Schölkopf et al. (2021); Sanchez & Tsaftaris (2022); Melistas et al. (2024) are considered a principal property for visual counterfactuals. In counterfactual generation a sub-

---

[1]Due to space constraints, we encourage the interested reader to refer to the Appendix A.5 for an explanation of the textual gradients computation.

[2]For simplicity and robustness, we employ the same LLM/VLM model (GPT-4) for all operations.

stantial challenge lies in incorporating the desired interventions (edits), while preserving unmodified other visual factors of variation which are not related to the assumed causal graph Monteiro et al. (2023) – a challenge closely tied to identity preservation of the observation (factual) Ribeiro et al. (2023). We evaluate counterfactual minimality in the text domain, offering a more interpretable alternative to conventional image-space metrics Zhang et al. (2018b). Specifically, we prompt a VLM to describe in detail both factual and counterfactual frames, excluding attributes associated with the assumed causal graph. We then embed the resulting descriptions using a BERT-based sentence transformer Wang et al. (2020) and compute their cosine similarity in the semantic space. The overall minimality metric can be expressed as follows:

$$\mathcal{P}_{min} = \text{"Describe this frame in detail, exclude DAG variables"}$$

$$Minimality(\mathcal{V}_{frame}, \mathcal{V}'_{frame}) = \cos\big(\tau_\phi(VLM(\mathcal{V}_{frame}, \mathcal{P}_{min})), \tau_\phi(VLM(\mathcal{V}'_{frame}, \mathcal{P}_{min}))\big) \quad (7)$$

where $\tau_\phi(.)$ denotes the semantic text encoder and $V_{frame}, V'_{frame}$ the factual and counterfactual frames.

# 5 EXPERIMENTS AND RESULTS

## 5.1 EVALUATION DATASET AND IMPLEMENTATION DETAILS

Following standard video editing evaluation protocols Wu et al. (2023b); Geyer et al. (2024); Cong et al. (2024); Liu et al. (2024a); Qi et al. (2023); Ku et al. (2024); Wang et al. (2025), we curated 67 text–video pairs from CelebV-Text Yu et al. (2023), an in-the-wild facial video dataset. For each video, we used the first 24 frames resized to $512 \times 512$ and assumed the data-generating process follows the causal graph in Figure 2 Yang et al. (2020); Melistas et al. (2024); Kladny et al. (2023).

We implement the parent prompt counterfactual function $g_{LLM}$ (Equation 2) with GPT-4, generating four counterfactual prompts per factual prompt by intervening on 'age,' 'gender,' 'beard,' and 'baldness' (Figure 2). For each prompt, we construct four multiple-choice questions targeting variables in the causal graph to assess causal effectiveness with the VLM (Equation 6).

The video counterfactual function $f_{LDM}$ (Equation 3) is implemented with three efficient T2I LDM-based video editing methods: FLATTEN (zero-shot, optical flow–guided attention for temporal coherence) Cong et al. (2024), Tune-A-Video (one-shot, fine-tuned spatio-temporal attention) Wu et al. (2023b), and TokenFlow (zero-shot, keyframe-based image editing with propagation) Geyer et al. (2024). We select these methods for their efficiency, while excluding cross-attention approaches such as Video-P2P Liu et al. (2024a) and FateZero Qi et al. (2023), which require identical source and edited prompt structures. All methods use Stable Diffusion v2.1 with DDIM sampling (50 steps) and classifier-free guidance (scale 4.5 for Tune-A-Video/TokenFlow, 7.5 for FLATTEN). The VLM counterfactual textual loss (Equation 4) is optimized with GPT-4 via TextGrad Yuksekgonul et al. (2025) (2 iterations). For evaluation, we use LLaVA-NeXT Li et al. (2024) for causal effectiveness (Equation 6) and GPT-4 Achiam et al. (2023) for minimality (Equation 7). All experiments are run on a single A100 GPU.

## 5.2 QUANTITATIVE EVALUATION.

We evaluate the generated counterfactual videos using metrics that capture key axiomatic properties of counterfactuals Galles & Pearl (1998); Halpern (2000), focusing on effectiveness Monteiro et al. (2023); Melistas et al. (2024) and minimality Melistas et al. (2024); Sanchez & Tsaftaris (2022). To assess visual fidelity and temporal coherence, we employ DOVER Wu et al. (2023a); Liu et al. (2024b), FVD Unterthiner et al. (2018), and CLIP Radford et al. (2021b) score between adjacent frames. We compare CSVC against vanilla video editing baselines using the initial counterfactual prompts, an LLM-based paraphrasing baseline where an LLM rephrases the target counterfactual prompt, and report results with and without the causal decoupling prompt.

From Table 1, observing the initial prompt rows, TokenFlow achieves the best trade-off between causal effectiveness and minimality among the baselines. Tune-A-Video generates effective counterfactuals but performs worst in terms of minimality across both LPIPS and the VLM-based metric. In terms of overall video quality and temporal consistency, TokenFlow and FLATTEN outperform Tune-A-Video, maintaining stronger visual coherence.

**Effectiveness.** To measure counterfactual effectiveness, we use VLMs prompted with multiple-choice questions on the intervened variables (age, gender, beard, bald). Table 1 reports VLM accuracy for each variable under these interventions. CSVC improves causal effectiveness across all baseline methods, with the highest scores achieved when incorporating the causal decoupling prompt (CSVC loss w/ causal decoupling), indicating better steering toward counterfactuals that break strong causal relations (e.g., adding a beard to a female). While naive LLM paraphrasing occasionally boosts gender interventions for FLATTEN and TokenFlow, it generally fails due to hallucinations or irrelevant content that the diffusion model cannot handle.

**Minimality.** To evaluate minimality, we use LPIPS Zhang et al. (2018b) and our proposed VLM-based metric (Equation 7). Our results reveal the trade-off between preserving proximity to the factual video and adhering to the counterfactual text conditioning. As shown in Table 1, LPIPS increases as counterfactual edits become more effective, with the VLM-based metric showing a similar trend through slight decreases in embedding cosine similarity. However, deviations from baseline methods remain marginal, indicating that CSVC achieves minimality scores comparable to vanilla frameworks while maintaining a balance with causal effectiveness.

**Video Quality and Temporal Consistency.** Table 1 reports quantitative results for video quality (DOVER, FVD) and temporal consistency (CLIP Radford et al. (2021b)). DOVER Wu et al. (2023a) shows only minor differences between baselines and our CSVC framework. FVD Unterthiner et al. (2018) increases slightly, reflecting greater deviation from the observational distribution as counterfactuals become more effective. CLIP-based temporal consistency remains close to the vanilla methods. Overall, our CSVC approach improves counterfactual effectiveness without compromising video realism or temporal coherence.

Table 1: Counterfactual Evaluation: Effectiveness, Minimality, Video Quality & Temporal Consistency.

| Method | Effectiveness (VLM Accuracy) | | | | Minimality | | Video Quality & Temp. Consistency | | |
|---|---|---|---|---|---|---|---|---|---|
| | age ↑ | gender ↑ | beard ↑ | bald ↑ | LPIPS ↓ | VLM-Min ↑ | DOVER ↑ | FVD ($\times 10^{-2}$) ↓ | CLIP-Temp ↑ |
| **FLATTEN** | | | | | | | | | |
| Initial Prompt | 0.597 | 0.746 | 0.313 | 0.418 | **0.161** | **0.791** | **0.841** | **3.472** | **0.982** |
| LLM Paraphrasing | 0.582 | 0.791 | 0.299 | 0.179 | 0.178 | 0.786 | **0.841** | 3.662 | **0.982** |
| CSVC w/o causal decoupling | 0.701 | 0.791 | 0.343 | 0.403 | 0.179 | 0.789 | 0.828 | 4.162 | 0.981 |
| CSVC w/ causal decoupling | **0.731** | **0.806** | **0.582** | **0.433** | 0.179 | 0.781 | 0.834 | 4.188 | **0.982** |
| **Tune-A-Video** | | | | | | | | | |
| Initial Prompt | 0.529 | **0.985** | 0.412 | 0.824 | **0.320** | **0.742** | 0.557 | **9.814** | **0.956** |
| LLM Paraphrasing | 0.507 | 0.970 | 0.433 | 0.358 | 0.396 | 0.695 | **0.596** | 13.581 | 0.939 |
| CSVC w/o causal decoupling | 0.779 | **0.985** | 0.426 | 0.868 | 0.362 | 0.722 | 0.552 | 11.600 | 0.955 |
| CSVC w/ causal decoupling | **0.824** | **0.985** | **0.676** | **0.912** | 0.370 | 0.717 | 0.558 | 11.840 | 0.955 |
| **TokenFlow** | | | | | | | | | |
| Initial Prompt | 0.672 | 0.836 | 0.388 | 0.522 | **0.227** | **0.776** | 0.787 | 7.712 | 0.984 |
| LLM Paraphrasing | 0.627 | 0.910 | 0.328 | 0.194 | 0.244 | 0.766 | **0.797** | **7.353** | 0.983 |
| CSVC w/o causal decoupling | 0.909 | **0.925** | 0.426 | 0.552 | 0.241 | 0.773 | 0.784 | 8.060 | 0.984 |
| CSVC w/ causal decoupling | **0.940** | 0.910 | **0.761** | **0.701** | 0.253 | 0.768 | 0.786 | 8.660 | **0.986** |

## 5.3 QUALITATIVE EVALUATION

Figure 3 shows qualitative results[3] across FLATTEN Cong et al. (2024), Tune-A-Video Wu et al. (2023b), and TokenFlow Geyer et al. (2024). The top row displays the factual video and prompt, while subsequent rows show counterfactuals generated with the initial counterfactual prompt, an LLM-paraphrased prompt, and our causally optimized prompt with CSVC. Our framework produces counterfactuals that accurately reflect the desired interventions, including breaking strong causal relationships (e.g., adding a beard to a woman), as well as causally faithful age and gender transformations. The results also showcase the effectiveness of CSVC over naive LLM prompt paraphrasing. Figure 4 illustrates CSVC with the FLATTEN method, where iterative gradient steps (2nd row) guide generation toward the intended intervention (youthful appearance), demonstrating controllable causal steering.

---

[3]Due to space constraints, additional qualitative results are provided in the Appendix A.8 and supplementary materials.

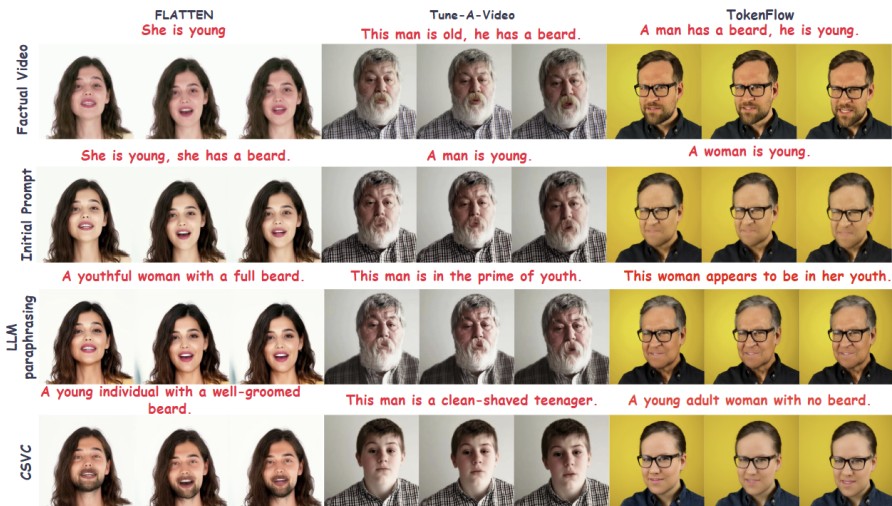

Figure 3: **Qualitative results**: **First panel:** intervention on beard (adding a beard to a woman: breaking strong causal dependencies). **Second panel:** intervention on age (making an old man with a beard appear young with no beard). **Third panel:** intervention on gender (transforming a man with a beard into a woman). The accuracy of the edits in the bottom row demonstrates the effectiveness of our CSVC framework in incorporating the assumed causal relationships.

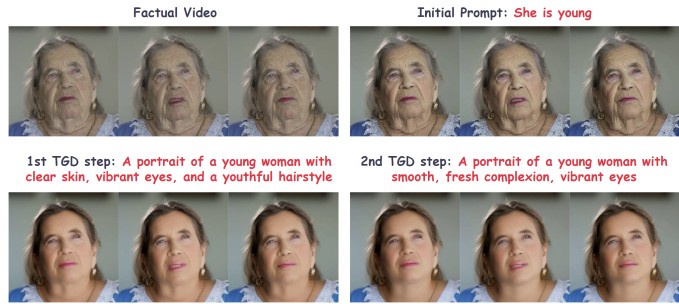

Figure 4: Counterfactual transformation of an elderly woman into a young woman (top row) TGD steps in the bottom row produced by our proposed CSVC with the FLATTEN Cong et al. (2024) editing method, which implements the counterfactual function $f_{LDM}$.

## 6 DISCUSSION AND LIMITATIONS

In this paper, we propose a causal framework, namely CSVC, for counterfactual video generation by implementing black-box counterfactual functions with generative AI models, where causal priors are encoded via target prompts that reflect relationships defined by a causal graph. CSVC enforces counterfactual conditioning by leveraging a VLM-based textual loss to iteratively refine the target counterfactual prompt, guiding the LDM toward generating novel OOD counterfactuals. This optimization strategy provides a principled approach to counterfactual generation, enhancing causal alignment while preserving visual realism, minimality, and temporal coherence. Experimental results highlight the effectiveness and controllability of CSVC, underscoring its potential to advance causal reasoning in large generative vision models. Importantly, our findings demonstrate that diffusion models can be effectively steered to generate OOD counterfactuals.

**Limitations.** We do not particularly add any loss to enforce temporal consistency beyond what each LDM baseline method does. It is quite possible that static interventions on the attributes could alter temporal consistency but we haven't observed it in our case. In video editing, the ability to manipulate temporal attributes such as actions or dynamic scenes is crucial. Constructing such graphs and datasets are necessary to develop and test such methods and are left for future work.

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

# A APPENDIX

## A.1 BLACK-BOX COUNTERFACTUAL FUNCTIONS

Counterfactual reasoning in structural causal models (SCMs) follows the abduction–action–prediction framework Pearl (2009). Given a factual variable $x = g(\epsilon, \text{pa})$ with parents pa and exogenous noise $\epsilon$, a counterfactual $x^*$ under intervened parents $\text{pa}^*$ is defined as $x^* = g(\epsilon, \text{pa}^*)$. This involves: (i) *abduction*, inferring $\epsilon$ from the factual observation; (ii) *action*, replacing pa with $\text{pa}^*$; and (iii) *prediction*, propagating the effect to obtain $x^*$. Since abduction is often non-invertible, the induced distribution over $\epsilon$ leads to multiple possible counterfactuals. To bypass explicit modeling of $\epsilon$, Monteiro et al. Monteiro et al. (2023) conceptualize SCM mechanisms as black-box counterfactual functions $f(x, \text{pa}, \text{pa}^*) \mapsto x^*$, which directly approximate counterfactual mappings.

## A.2 EVALUATION DATASET

We curated an evaluation dataset consisting of 67 text-video pairs sourced from the large-scale facial text–video dataset CelebV-Text Yu et al. (2023). We extracted the first 24 frames from each video and resized them to a resolution of $512 \times 512$. Each video in CelebV-Text is associated with a text prompt describing static appearance attributes. We model the data-generating process using the causal graph shown in Figure 5. Given the factual (original) text prompt for each video, sourced from CelebV-Text Yu et al. (2023), we derive four counterfactual (target) prompts that are as similar as possible to the factual prompt, differing only in the specified interventions. To produce the counterfactual prompts and incorporate the interventions, we follow the assumed causal relationships depicted in the causal graph (Figure 5)–for example, older men are more likely to have a beard or be bald than younger men, while women typically do not exhibit facial hair or baldness.

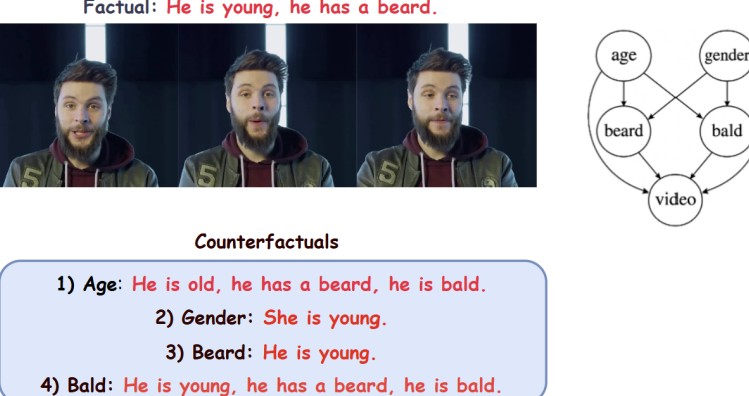

Figure 5: Evaluation dataset structure: Each factual prompt, sourced from CelebV-Text, is associated with four counterfactual prompts. Each counterfactual (target) represents an intervention on one of the following variables–age, gender, beard, or baldness. Interventions on upstream causal variables (e.g., age or gender) may lead to changes in downstream variables (e.g., beard or baldness), which are automatically incorporated into the counterfactual prompt.

## A.3 ADDITIONAL IMPLEMENTATION DETAILS

For each baseline video editing method (FLATTEN Cong et al. (2024), Tune-A-Video Wu et al. (2023b), and TokenFlow Geyer et al. (2024)), we adopt the default experimental hyperparameters provided in the original works. In our experiments, we implement the VLM-based textual loss in our CSVC framework using the GPT-4o model via the OpenAI API. However, our approach is also compatible with local VLMs currently supported by the TextGrad package Yuksekgonul et al. (2025). The LLM used to perform the TextGrad update (Equation 5) is GPT-4–the same model used for the VLM loss. We also use the GPT-4o API to compute the VLM minimality metric, as it offers improved filtering of the causal graph variables in the generated text descriptions. In addition, for

the BERT-based semantic text encoder $\tau_\phi$ used in Equation 7 to generate semantic text embeddings, we leverage the *all-MiniLM-L6-v2* model Wang et al. (2020), which maps the text descriptions into a 384-dimensional vector space. Lastly, to evaluate effectiveness as expressed in Equation 6, we utilize the *llava-hf/llava-v1.6-mistral-7b-hf*

## A.4 PROMPTS

### A.4.1 LLM MECHANISM: IN-CONTEXT LEARNING PROMPT

In Listing 1, we provide a part of the GPT-4 in-context learning prompt used to derive the initial counterfactual parent prompts from the factual prompts for each video by incorporating the causal graph (Figure 2). To generate the 4 counterfactual prompts per video, we additionally supply GPT-4 with all 67 factual descriptions of the original videos. In total, we produce 268 ($67 \times 4$) counterfactual prompts (four per video). The full prompt is included in our code.

**Listing 1:** LLM in-context learning prompt $P_{ICL}$

```
You are given a causal DAG with 4 variables: age, gender, beard, and baldness.

Causal relationships:
- age -> beard
- age -> bald
- gender -> beard
- gender -> bald

Domain knowledge:
1. Older men are more likely to have a beard and be bald compared to younger men.
2. Men are more likely to have a beard and be bald compared to women.

Task:
Given a factual prompt that describes a person (e.g., He is young, he has a beard),
generate 4 counterfactual prompts by intervening on each variable (age, gender, beard, bald) w
Examples:
---
Factual:
He is young
Counterfactuals:
age: He is old, he has a beard, he is bald
gender: She is young
beard: He is young, he has a beard
bald: He is young, he is bald
---
Factual:
He is young, he has a beard
Counterfactuals:
age: He is old, he has a beard, he is bald
gender: She is young
beard: He is young
bald: He is young, he has a beard, he is bald
---
Factual:
He is old, he is bald
Counterfactuals:
age: He is young
gender: She is old
beard: He is old, he has a beard, he is bald
bald: He is old
---
Factual:
She is old
Counterfactuals:
age: She is young
gender: He is old, he has a beard, he is bald
beard: She is old, she has a beard
```

```
bald: She is old, she is bald
```

### A.4.2 Evaluation Instruction

We outline the methodology used to construct the evaluation instruction prompt for the VLM-based textual loss of the CSVC framework, as described in Section 4.1. First, given the factual (source) prompt of the original video and the initial counterfactual (target) prompt–we programmatically extract the target interventions by comparing the two. In Listing 2, we provide representative examples.

**Listing 2:** Target Interventions Extraction

```
Factual prompt: This woman is young.
Initial Counterfactual prompt: This woman is old.
Target interventions: old (age)

Factual prompt: He is young, he has a beard.
Initial Counterfactual prompt: She is young.
Target interventions: woman, no-beard (gender)

Factual prompt: This woman is young.
Initial Counterfactual prompt: This woman is young, she has a beard.
Target interventions: beard (beard)

Factual prompt: A man is young.
Initial Counterfactual prompt: A man is young, he is bald.
Target interventions: bald (bald)
```

Given the initial counterfactual prompt and the target interventions, we provide the VLM with the following evaluation instruction:

**Listing 3:** VLM Evaluation Instruction

```
You are given an image of a person's face.

- A counterfactual target prompt is provided: {counterfactual_prompt}

- Corresponding interventions are specified: {target_interventions}

- Evaluate how well the given image aligns with the specified
    counterfactual attributes in the target prompt.

- Calculate an accuracy score based only on the attributes that were
    explicitly modified (i.e., the interventions).

- Do not describe or alter any other visual elements such as expression,
    hairstyle, background, clothing, or lighting.

- Identify and list any attributes from the interventions that are
    missing or incorrectly rendered.

- Criticize.

- Suggest improvements to the counterfactual prompt to better express
    the intended interventions.

- The optimized prompt should maintain a similar structure to the
    original prompt.

- If the alignment is sufficient, return: "No optimization is needed".
```

### A.4.3 CAUSAL DECOUPLING PROMPT

We further augment the evaluation instruction prompt with a causal decoupling prompt (Listing 4), in cases where interventions involve downstream variables (e.g., beard, bald) in the causal graph. This results in optimized prompts that exclude references to upstream variables (e.g., age, gender), effectively breaking the assumed causal relationships and simulating graph mutilation Pearl (2009). By using such prompts, the LDM backbone of the video editing method can generate OOD videos that violate the assumptions of the causal graph–for example, by adding a beard to a woman.

**Listing 4:** Causal Decoupling Prompt

```
If either beard or bald appears in target_interventions, do not include
    references to age or gender.
```

### A.5 EVALUATIVE TEXTUAL FEEDBACK FROM VLM-BASED LOSS AND TEXTUAL GRADIENT COMPUTATION

For demonstration purposes, we provide the textual feedback from the VLM-based loss in our CSVC framework during counterfactual parent prompt refinement for the first video in Figure 1 (transforming an old woman into a young one) with the TokenFlow Geyer et al. (2024) editing method. In addition, we present the corresponding textual gradient $\frac{\partial \mathcal{L}}{\partial \mathcal{P}}$, which is used to update the initial prompt via the TextGrad Yuksekgonul et al. (2025). First, we generate the counterfactual video using the initial counterfactual prompt (A woman is young), which represents an intervention on the age variable. Then, we provide a generated counterfactual frame to the VLM for evaluation.

**Factual Video**

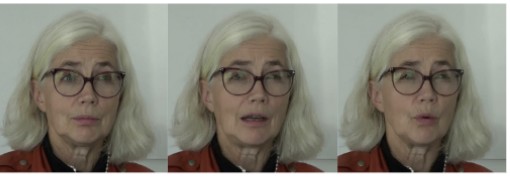

**Initial Prompt: A woman is young**

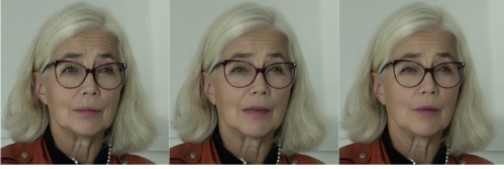

Figure 6: Counterfactual video generated using the initial counterfactual parent prompt, which failed to incorporate the target intervention (young).

In Listing 5, we present the textual feedback produced by the VLM given a frame from the ineffective generated counterfactual video of Figure 6.

**Listing 5:** VLM textual feedback from the first prompt refinement iteration

```
  The image does not align well with the specified counterfactual
     attributes from the target prompt. The intervention specified was
     "young," but the person depicted in the image appears to be an
     older adult, likely in her 60s or 70s, based on visible signs of
     aging such as wrinkles and grey hair.

**Accuracy Score:** 0/1 (The attribute "young" was not correctly
    rendered)

**Failed Attributes:**
- Youthfulness: The individual in the image does not appear young.
```

```
**Suggested Improvement to the Counterfactual Prompt:**
"A woman in her early twenties"

This optimized prompt more specifically targets the desired age range,
    enhancing the likelihood that the generated image will meet the
    counterfactual criteria.
```

In addition, in Listing 7, we display the textual gradients $\frac{\partial \mathcal{L}}{\partial \mathcal{P}}$, which contain detailed criticisms and suggestions for improving the initial counterfactual parent prompt based on the VLM's textual feedback. To compute the textual gradient $\frac{\partial \mathcal{L}}{\partial \mathcal{P}}$, TextGrad leverages a backward engine (LLM/VLM) that is prompted with an augmented prompt derived by combining internal TextGrad system prompts, the evaluation instruction (Listing 3), the generated counterfactual frame, the refined counterfactual prompt $\mathcal{P}$, and the evaluative textual feedback (Listing 5). In Listing 6 we present part of the internal prompt that TextGraD employs to produce textual gradients.

**Listing 6:** TextGrad Internal Prompt for Computing $\frac{\partial \mathcal{L}}{\partial \mathcal{P}}$

```
  You will give feedback to a variable with the following role: <ROLE>
      prompt to optimize </ROLE>.
Here is an evaluation of the variable using a language model:

Above messages are the <LM_INPUT>

<LM_SYSTEM_PROMPT>
You are an evaluation system that evaluates image-related questions.
</LM_SYSTEM_PROMPT>

<LM_OUTPUT>
The image does not align well with the specified counterfactual
    attributes from the target prompt. The intervention specified was
    "young," but the person depicted in the image appears to be an older
    adult, likely in her 60s or 70s, based on visible signs of aging
    such as wrinkles and grey hair.

**Accuracy Score:** 0/1 (The attribute "young" was not correctly
    rendered)

**Failed Attributes:**
- Youthfulness: The individual in the image does not appear young.

**Suggested Improvement to the Counterfactual Prompt:**
"A woman in her early twenties"

This optimized prompt more specifically targets the desired age range,
    enhancing the likelihood that the generated image will meet the
    counterfactual criteria.
</LM_OUTPUT>

<OBJECTIVE_FUNCTION>
Your goal is to give feedback and criticism to the variable given the
    above evaluation output.
Our only goal is to improve the above metric, and nothing else.
</OBJECTIVE_FUNCTION>

We are interested in giving feedback to the prompt to optimize for this
    conversation.
Specifically, give feedback to the following span of text:

<VARIABLE> A woman is young </VARIABLE>

Given the above history, describe how the prompt to optimize could be
    improved to
improve the <OBJECTIVE_FUNCTION>. Be very creative, critical, and
```

**Listing 7:** Textual gradient $\frac{\partial \mathcal{L}}{\partial \mathcal{P}}$

```
The variable "A woman is young" lacks specificity and clarity, which
    likely contributed to the misalignment between the generated image
    and the intended attribute of youthfulness. The term "young" is
    highly subjective and can be interpreted variably depending on
    cultural, personal, and contextual factors. This ambiguity in the
    prompt may have led the Text-to-Image model to generate an image
    that does not meet the counterfactual criteria specified.

**Feedback and Criticism:**

1. **Specificity in Age Description**: The term "young" should be
    replaced with a more precise age or age range to guide the model
    more effectively. For instance, specifying "a woman in her early
    twenties" or "a woman aged 20-25 years" would provide a clearer
    target for the model, reducing ambiguity and increasing the
    likelihood of generating an image that aligns with the youthful
    attribute desired.

2. **Inclusion of Contextual Details**: While the prompt should remain
    focused on the age attribute, adding minimal contextual details
    that align with youthful appearances could enhance the
    effectiveness of the prompt. For example, mentioning "a youthful
    demeanor" or "fresh-faced" could cue the model to focus on features
    typically associated with younger individuals.

3. **Avoiding Overly Broad Descriptors**: The use of broad descriptors
    like "young" without additional qualifiers can lead to a wide range
    of interpretations. It's beneficial to avoid these where precision
    is required for generating specific outcomes.

4. **Testing and Iteration**: After adjusting the prompt, it would be
    advisable to test the new prompt with the model to see if the
    changes yield a more accurate representation. This iterative
    approach can help fine-tune the prompt based on actual outcomes.

By addressing these points, the prompt "A woman is young" can be
    optimized to more effectively communicate the desired attribute of
    youthfulness to the Text-to-Image model, thereby improving the
    alignment of the generated image with the counterfactual target.
```

The textual gradients $\frac{\partial \mathcal{L}}{\partial \mathcal{P}}$ (Listing 7) are provided as input to Textual Gradient Descent Yuksekgonul et al. (2025), which leverages an LLM to update the optimized variable (prompt), as described in Equation 5. For simplicity and robustness in our experiments, we use the same LLM/VLM (GPT-4) for all operations: producing textual evaluative feedback, computing textual gradients, and updating the prompt with Textual Gradient Descent. After the TGD update the counterfactual prompt becomes: *A woman in her early 20s with vibrant expression*.

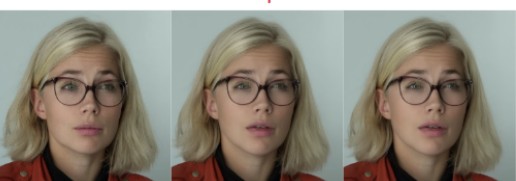

Figure 7: Counterfactual video generated using the refined counterfactual parent prompt, which successfully incorporates the target intervention (young).

In Listing 8, we display the textual feedback from the VLM after providing it with a frame from the effective counterfactual video generated using the optimized prompt (Figure 7). With this prompt,

the age intervention (young) is successfully incorporated. Consequently, the VLM returns a "no optimization" response, and the prompt optimization process terminates.

**Listing 8:** VLM feedback from the second counterfactual prompt refinement iteration

```
The input frame aligns well with the specified counterfactual attribute
    of appearing "young." The individual in the image presents as a
    young adult, which matches the intervention target of portraying
    youth. Therefore, the accuracy score based on the attribute of
    appearing young is high.

No attributes from the interventions failed to appear or were
    incorrectly rendered in this context.

Since the image successfully aligns with the desired attribute of youth,
    there is no need for optimization of the prompt. The response is
    "no_optimization".
```

### A.6 VLM-BASED METRICS FOR ASSESSING EFFECTIVENESS AND MINIMALITY

#### A.6.1 EFFECTIVENESS

We present the VLM pipeline for evaluating causal effectiveness. As shown in Figure 10, the VLM receives as input the generated counterfactual frame and a multiple-choice question–extracted from the counterfactual prompt that corresponds to the intervened attribute. Since we edit static attributes, a single frame is sufficient to assess the effectiveness of the interventions. An accuracy score is calculated across all generated counterfactual frames for each intervened variable (age, gender, beard, baldness) (Equation 6).

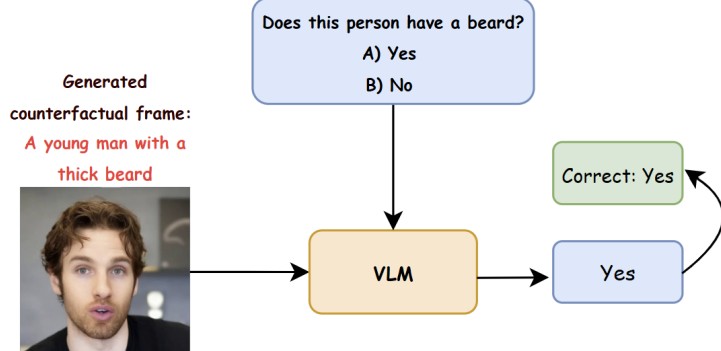

Figure 8: VLM causal effectiveness pipeline: example of a beard intervention.

#### A.6.2 MINIMALITY

In Figure 9, we showcase the VLM pipeline for evaluating minimality (Equation 7). The VLM takes as input frames extracted from the factual and counterfactual videos and produces text descriptions that exclude attributes from the causal graph. These text descriptions are then passed through a BERT-based semantic encoder Wang et al. (2020) to generate semantic embeddings. The final minimality score is computed as the cosine similarity between these embeddings. The exact prompt used to instruct the VLM to filter the text descriptions from the causal graph variables is provided in Listing 9.

**Listing 9:** VLM Minimality Prompt

```
Remove any references to age, gender (man, woman, he, she), beard, hair
    (including hairstyle, color, style, and facial hair), and baldness
    from the description.

Return only the filtered version of the text, without commentary or
    formatting.
```

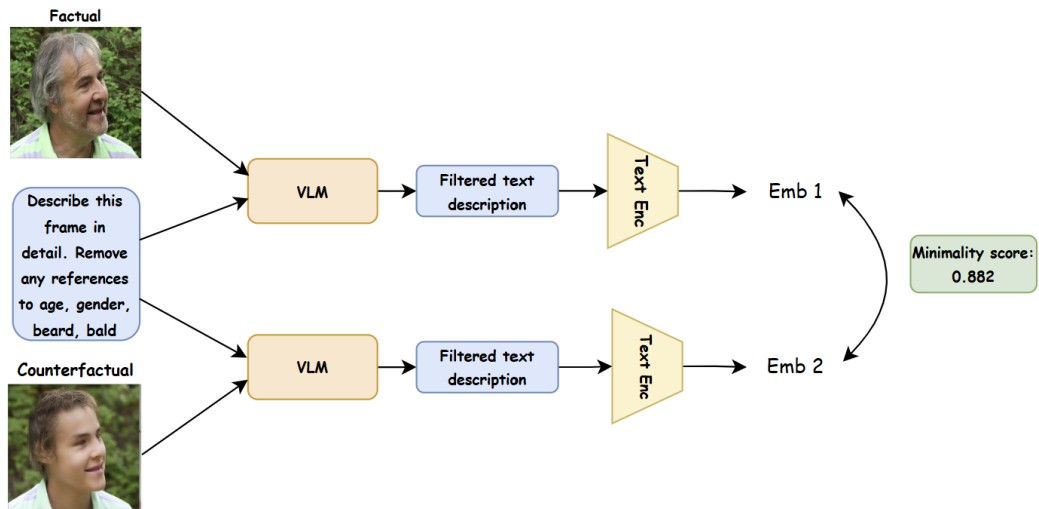

Figure 9: VLM minimality pipeline: example of a gender intervention.

In Figure 10, we display the filtered text descriptions produced by the VLM. This specific factual and counterfactual pair achieves a VLM minimality score of 0.882. We observe that by measuring the semantic similarity of the VLM-generated text descriptions, we can isolate factors of variation not captured by the causal graph and effectively measure their changes under interventions on the causal graph variables.

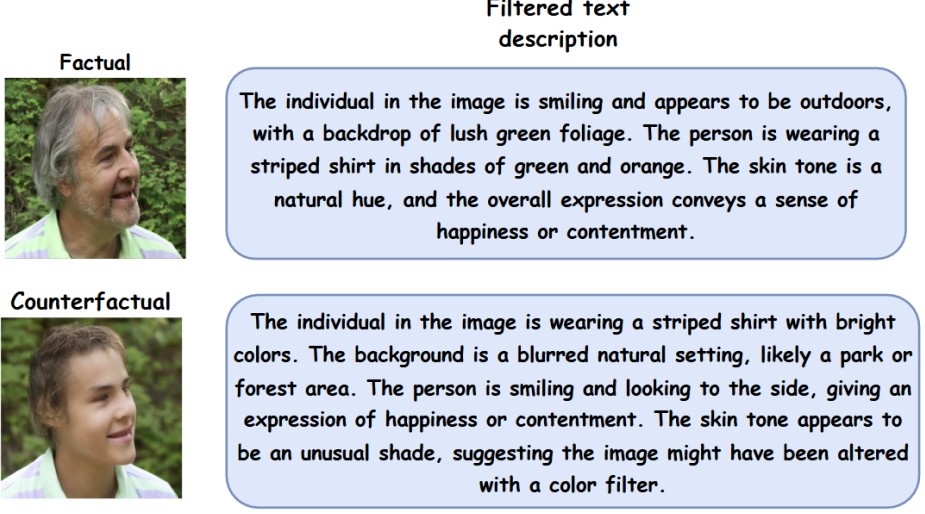

Figure 10: Filtered text descriptions derived from the VLM

## A.7 BROADER IMPACT

Our framework (CSVC) for generating causally faithful video counterfactuals enhances video synthesis, interpretable AI, and content manipulation by providing better controllable edits. This could improve automated content generation in fields like healthcare (e.g., simulating treatment outcomes or disease progression under varied causal conditions), education (e.g., allowing students to observe video counterfactuals of complex processes, such as surgical procedures or engineering designs), and digital media (e.g., enabling creative content manipulation). Furthermore, it can potentially address ethical concerns, regarding thoroughly evaluating the misuse of deepfake technologies, highlighting the need for responsible guidelines and safeguards.

## A.8 MORE QUALITATIVE RESULTS

In Figures 11, 12, 13, 14, and 15, we present additional qualitative results generated with our proposed framework, Causal Steering for Video Counterfactuals (CSVC), using different LDM-based video editing systems to implement the black-box video counterfactual function.

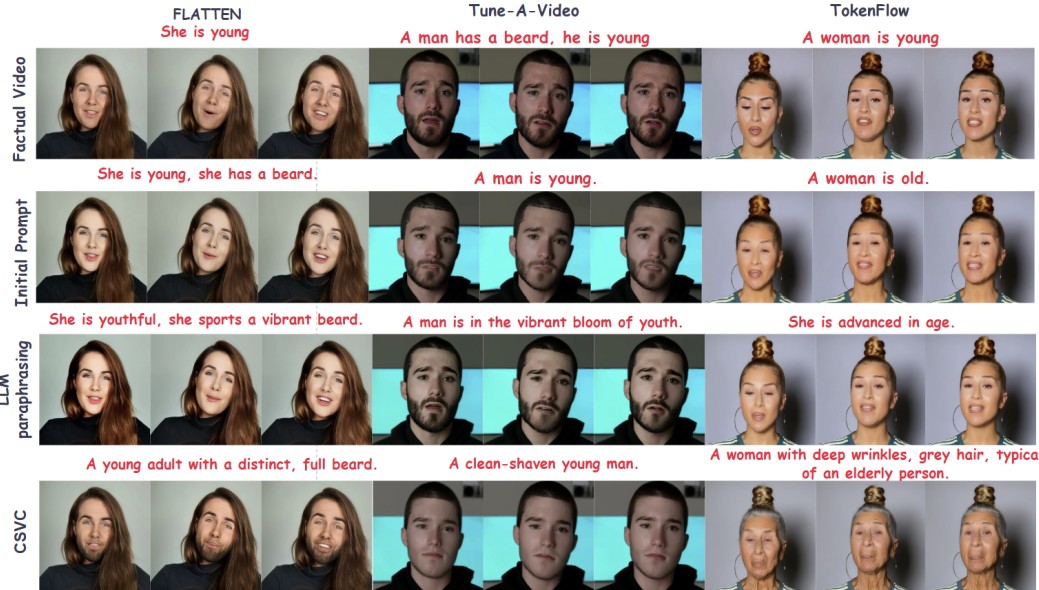

Figure 11: **Qualitative results**: Generated counterfactual videos illustrate the positive effect of our proposed CSVC framework (bottom row) when applied to recent video editing systems (FLATTEN Cong et al. (2024), Tune-A-Video Wu et al. (2023b), and TokenFlow Geyer et al. (2024)). **First panel:** intervention on beard (adding a beard to a woman). **Second panel:** intervention on beard (removing a beard from a man). **Third panel:** intervention on age (aging a woman).

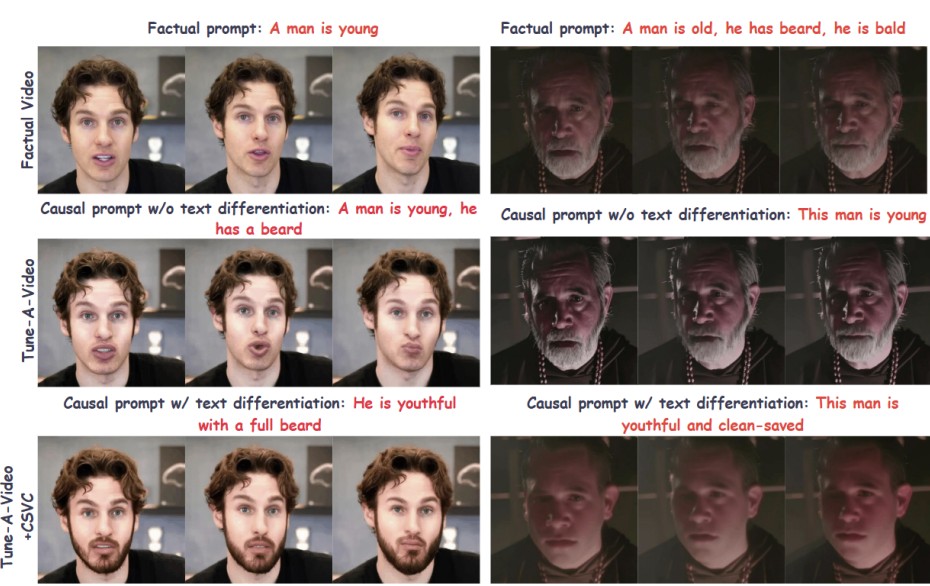

Figure 12: First panel: intervention on beard. Second panel: intervention on age.

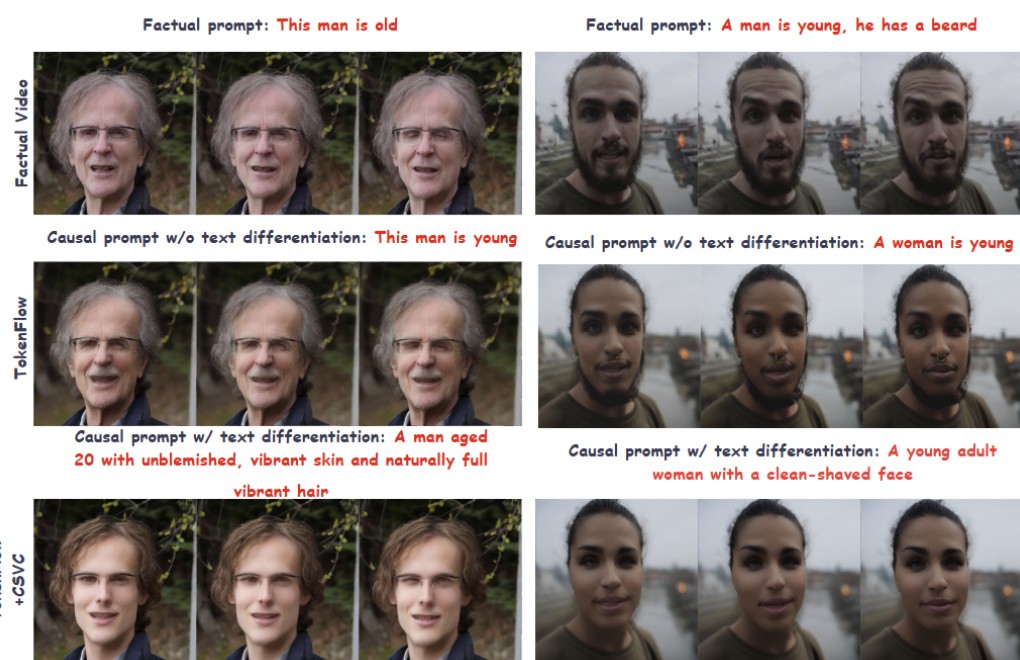

Figure 13: First panel: intervention on age. Second panel: intervention on gender.

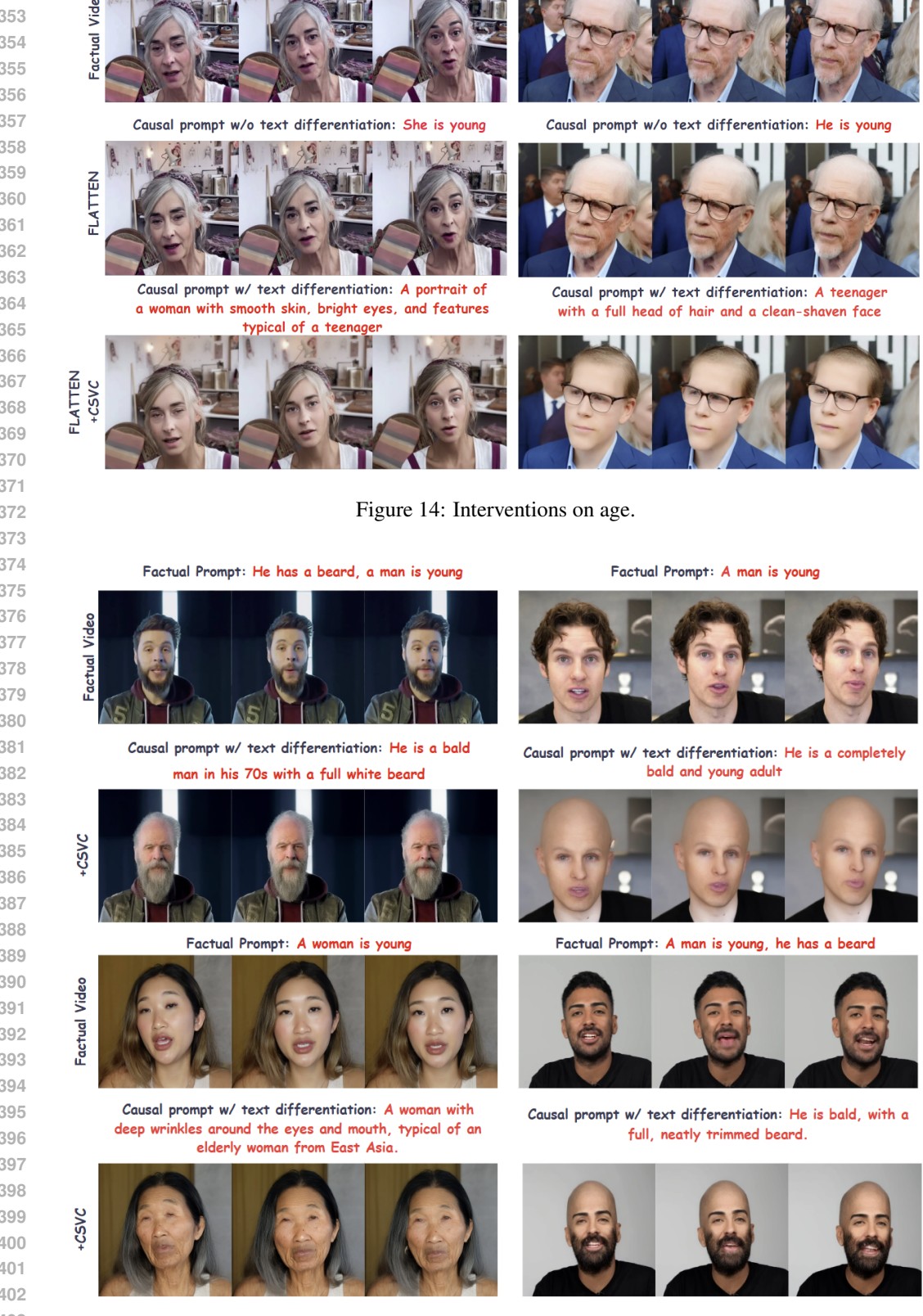

Figure 14: Interventions on age.

Figure 15: First panel: Interventions on age. Second panel: Interventions on baldness

