# OpenReview forum: "Causally Steered Diffusion for Video Counterfactual Generation"
_ICLR.cc/2026/Conference — ICLR 2026 Conference Withdrawn Submission_

### Official Review · Reviewer_Tk7y · 2025-10-31

**Soundness:** 2
**Presentation:** 2
**Contribution:** 2
**Rating:** 2
**Confidence:** 3

**Summary:**

This paper proposes a framework for generating video counterfactuals using only black-box models. Observing that video editors often ignore prompts that contradict the source video , the authors introduce a "VLM-based textual loss". This mechanism uses a VLM to evaluate the generated video and provide "textual feedback" (criticism). This feedback iteratively refines the input prompt, "steering" the black-box editor to produce a causally faithful result without any fine-tuning.

**Strengths:**

+ The framework's core novelty is its use of a VLM as a "textual loss" to create a closed-loop, black-box optimization system for causal steering
+ This black-box approach can be applied to any LDM-based video editor without requiring model access or fine-tuning

**Weaknesses:**

- The paper claims it overcome the shortage of existing work but fails to quantitatively compare against any published work. The evaluation is limited to its own weak baselines.
- The only baseline, "LLM Paraphrasing", is completely undefined. The paper provides no prompt, methodology, or strategy for "LLM Paraphrasing" baseline, rendering it unverifiable.
- The paper's "video" contribution is overstated . The method only intervenes on static appearance attributes and explicitly does not handle temporal attributes like actions .
- All experiments are confined to a single, trivial causal graph of just four facial attributes (age, gender, beard, baldness), offering no evidence that this complex method scales beyond this narrow use case.

**Questions:**

1. Please provide the comparisons against existing published works you mentioned in related work.
2. Could you provide a detailed explanation of the baseline implementation for “LLM Paraphrasing” and analyze why LLM Paraphrasing itself cannot achieve the same results?
3. Could we conduct experiments on a broader scale to test more counterfactual scenarios? Relying solely on the influence of age and gender seems too easy. The generalizability remains unclear.
4. Could you explain what makes this task unique in the context of video scenes, since it doesn't seem to involve the core distinction between video and image?

---

### Official Review · Reviewer_gSfz · 2025-10-31

**Soundness:** 3
**Presentation:** 3
**Contribution:** 2
**Rating:** 4
**Confidence:** 4

**Summary:**

The paper proposes a counterfactual video editing process using multiple pretrained foundation models to engineer causality-aware text prompts for video editing.
The core idea is to embed Pearl-style counterfactual ideas to create meta prompts that enable causality-aware edits to a video editing prompt.
The paper does this by iteratively refining an editing text prompt given the causal graph, some in-context-learning examples and some meta instructions.
The paper then demonstrates the method empirically both quantitatively and qualitatively on real human face videos.

**Strengths:**

- Introduces a simple prompting method with causal-awareness to produce better counterfactual prompts for video generation (a causal-aware type of prompt engineering).
- The paper is well-written and easy to understand.
- The paper appropriately leverages pretrained models and does not add unnecessary complexity to the methods.
- Introduce some simple VLM-based metrics for counterfactual video generation evaluation.

**Weaknesses:**

- The paper is an application-focused paper since it composes multiple other models to accomplish it's objectives. From an application standpoint, this is a strong and well-explained approach that doesn't do anything unnecessary and uses the natural tools to accomplish the goal. From a fundamental side, it does not make a significant contribution since even the optimization is based on TextGrad. Thus, the methodological novelty is relatively low---though this is not necessarily a rejection-worthy concern given the application focus.

- This method does require a causal graph to be known a priori. This should be noted as a key limitation for more general causal video editing.

- One potential issue with the proposed method is that the LLMs or VLMs may be using implicit world knowledge rather than actually using the provided causal graph. Currently, you only use a "realisitic" causal graph that would align with the pretrained knowledge of foundation models. Thus, it is unclear whether the editing ability comes from the provided causal graph or simply world knowledge.
   - To more carefully explore your method, I would be interested in another experiment with a non-realistic causal graph (e.g., one that is opposite of true causal factors). Would your method still obey the (incorrect) causal graph? This would be an interesting way to stress test whether it is actually using your causal graph or just using world knowledge.

- Experimental baselines are limited
   - Because this is primarily prompt engineering, I feel that only having a few baselines is a little odd. You have an initial prompt and LLM paraphrasing as the two main baselines. However, it seems there should be other baselines like a meta-prompt or thinking-based prompt enhancement or other types of prompt editing techniques. For example, first describe the video verbosely and then ask it to causally change a part of the prompt to create a complex causally modified prompt.
   - Needless to say, more prompt engineering baselines seem appropriate. What is the best zero-shot prompt engineering technique that can leverage VLMs first to get textual respresentation of video and then the target modification.

- (Minor/Typo) Please use \citep and \citet macros correctly. If taking out a citation leaves the sentence structure intact, you should use \citep. If the citation is needed for the sentence, then you should use \citet.

(I would probably increase my score if you could demonstrate that your method actually respects the input causal graph, which could differ from reality or differ from pretrained knowledge (like a near inverse of the given causal graph). This would show it's not just using pretrained knowledge but actually making edits based on the given causal graph.)

**Questions:**

- Why do you remove all causal DAG variables from minimality? It seems you should only exclude intervention and downstream variables, not upstream variables. In fact, upstream variables should be preserved.

**Details Of Ethics Concerns:**

- This paper on counterfactual video editing definitely needs some ethical discussion that is missing from the current paper.
- Possible issue could include impersonating people or editing them older or younger without their knowledge.
- There could also be ethical issues with changing someone from an adult to a minor.
- The ethical questions are more intense because videos can be more convincing than other forms of media like images.
- Another concern is modifying gender of a factual person against their knowledge.
- There seems to be many ways that this could be harmful or unethical so a strong discussion section seems critical.

---

### Official Review · Reviewer_mBFK · 2025-11-01

**Soundness:** 3
**Presentation:** 3
**Contribution:** 3
**Rating:** 6
**Confidence:** 3

**Summary:**

This paper introduces a causal-steering framework for counterfactual video generation that treats text-guided diffusion editors as black-box counterfactual mechanisms and uses LLMs to transform factual prompts into causally consistent counterfactual parent prompts defined by a DAG. To enforce the intended interventions, it proposes a VLM-based textual loss optimized via Textual Gradient Descent that iteratively refines the prompt using “textual gradients,” requiring no access or fine-tuning of the underlying diffusion models and remaining compatible with any black-box editor; evaluated on real-world facial videos.

**Strengths:**

- The paper proposes a causal-steering framework that is plug-and-play with any black-box, text-guided diffusion video editor.

- It introduces a VLM-based textual loss optimized via Textual Gradient Descent, which directly refines counterfactual parent prompts to enforce the intended interventions without accessing or fine-tuning the editor’s internal parameters.

- The evaluation is principled and multifaceted.

**Weaknesses:**

- The evaluation is based on a small, narrow benchmark. It contains only 67 clips curated from CelebV-Text with static facial attributes.

- The paper does not report variability or statistical significance for Table 1.

- The paper provides no public link to code or the curated evaluation split.

**Questions:**

- Could you report standard deviations for all metrics?

- Several gains look large, but where improvements are less than 3–5 points, can you confirm they remain significant?

- Can you isolate the contribution of each component? (i.e., ablation studies)

- What are per-clip wall-clock times and GPU memory for your pipeline vs. each baseline, including the cost of textual-gradient optimization?

---

### Official Review · Reviewer_vXJD · 2025-11-01

**Soundness:** 2
**Presentation:** 2
**Contribution:** 1
**Rating:** 2
**Confidence:** 4

**Summary:**

This paper proposes CSVC (Causal Steering for Video Counterfactuals), a prompt-engineering framework that combines structural causal models (SCMs) with latent diffusion models (LDMs) for generating video counterfactuals.
The approach uses LLMs to generate counterfactual text prompts given a predefined causal graph, LDM-based video editors to generate corresponding videos, and a VLM-based textual loss (optimized via TextGrad) to refine prompts.
The framework is evaluated on 67 text-video pairs from CelebV-Text.

**Disclaimer**: I don't have much experience in video editing, and therefore I will focus on evaluating the causal foundation and rigor of their methodology and might miss understanding of comparisons with existing literature.

**Strengths:**

1. Applying causal reasoning to video generation is valuable and underexplored.
2. The framework in Figure 2 is practical and easy to implement. It might be useful to help improve video-editing models (though I have doubts about whether it would really work as will be explained below).
3. Despite some structural issues (why put 4.2 and 4.3 in methodology section?), the paper is overall easy to follow.

**Weaknesses:**

**Causality and positioning**
1. The causal framework is overclaimed. The text is saturated with SCM/causality-related language. However, the causal graph has a minimal role as far as I understand. It is essentially used as an input to the LLM to help generate the counterfactual prompt, which has minimal interaction with the LDM.
2. Following this concern, I wonder what would happen if we did not include the causal graph and just let the LLM edit the prompt, keeping everything else the same.
3. The framework assumes access to a causal graph, which limits its applicability.
4. Line 97 – the authors claim that "this stands in contrast to approaches based on attention engineering, which offer suboptimal solutions." I don't understand why those methods offer suboptimal solutions compared to the proposed method. The authors fail to justify this either from a theoretical perspective or an empirical perspective.

**Other technical concerns**
1. The authors use a VLM to compute "textual loss" to update the prompt. The authors also use VLMs for final evaluations. Is the same VLM used in both cases? If so, I think the comparison is not fair.
2. I don't understand why the proposed framework would help with temporal coherence. Essentially, the counterfactual video is generated using an edited prompt. I am not sure why this helps with temporal consistency.
3. Minimality is proposed for visual counterfactuals. However, the authors measured it in the text domain.

**Empirical study**
1. Could the authors explain why they exclude comparison against Video-P2P and FateZero. They claim that those methods require identical source and edited prompt structures, but I don't understand why that is a problem.
2. The dataset seems small-scale and limited. Why don't the authors compare on the datasets the baselines used (DAVIS)?

**Questions:**

The current citation style makes it hard to distinguish normal text and citations.

---

### Note · Authors · 2025-11-15

I have read and agree with the venue's withdrawal policy on behalf of myself and my co-authors.